# Effect of Polycyclic Aromatic Hydrocarbons Exposure on Cognitive Development in 5 Years Old Children

**DOI:** 10.3390/brainsci10090619

**Published:** 2020-09-07

**Authors:** Barbora Blazkova, Anna Pastorkova, Ivo Solansky, Milos Veleminsky, Milos Veleminsky, Katerina Urbancova, Veronika Vondraskova, Jana Hajslova, Jana Pulkrabova, Radim J. Sram

**Affiliations:** 1Faculty of Health and Social Sciences, University of South Bohemia, 370 05 Ceske Budejovice, Czech Republic; barbora.blazkova01@gmail.com (B.B.); PastorkovaA@seznam.cz (A.P.); solansky.ivo@seznam.cz (I.S.); veleminsky@volny.cz (M.V.J.); veleminsky@zsf.jcu.cz (M.V.); 2Institute of Experimental Medicine CAS, 142 20 Prague, Czech Republic; 3Hospital Ceske Budejovice, a.s., 370 01 Ceske Budejovice, Czech Republic; 4Faculty of Food and Biochemical Technology, University of Chemistry and Technology, 166 28 Prague, Czech Republic; urbancova.k@gmail.com (K.U.); veronika.vondraskova@vscht.cz (V.V.); jana.hajslova@vscht.cz (J.H.); jana.pulkrabova@vscht.cz (J.P.)

**Keywords:** polycyclic aromatic hydrocarbons, OH–PAH metabolites in urine, psychological tests, cognitive development, Bender Visual Motor Gestalt Test, Raven Colored Progressive Matrices

## Abstract

**Objectives**: To analyze the impact of polycyclic aromatic hydrocarbons (PAHs) in ambient air at the time of delivery and five years of age on cognitive development in five year old children. **Materials and Methods**: Two cohorts of children born in the years 2013 and 2014 from Karvina (Northern Moravia, *n* = 70) and Ceske Budejovice (Southern Bohemia, *n* = 99) were studied at the age of five years for their cognitive development related to the exposure to PAHs, determined in the ambient air as the concentration of benzo[a]pyrene (B[a]P) and OH–PAH (hydroxy-PAH) metabolites in urine of the newborns at the time of delivery. As psychological tests, the Bender Visual Motor Gestalt Test (BG test) and the Raven Colored Progressive Matrices (RCPM test) were used. **Results**: Concentrations of B[a]P in the third trimester of mother’s pregnancies were 6.1 ± 4.53 ng/m^3^ in Karvina, and 1.19 ± 1.28 ng/m^3^ (*p* < 0.001) in Ceske Budejovice. Neither the outcome of the RCPM test nor the BG test differed between children in Karvina vs. Ceske Budejovice, or boys vs. girls. Cognitive development in five year old children was affected by the higher exposure to PM2.5 during the third trimester in girls in Karvina. **Conclusions**: We did not observe any significant effect of prenatal PAH exposure on psychological cognitive tests in five year old children.

## 1. Introduction

Already thirty years ago, Sram [1] hypothesized that air pollution exposure of the fetus developing in uterus may induce functional changes in the nervous system, which may later be expressed as developmental disorders or neurobehavioral impairment.

The first report of the behavioral effects of benzo[a]pyrene (B[a]P) exposure in rats was published by Saunders et al. in 2001 [2].

The effect of prenatal exposure to airborne polycyclic aromatic hydrocarbons (PAHs) on neurodevelopment was studied by Frederica Perera in New York for a long-term Columbia University cohort of non-smoking African-American and Dominican mothers and children. The first results indicated DNA damage and impaired fetal growth [3].

This cohort was followed at the age of three years by the Bayley test [4]. Cognitive and psychomotoric development was evaluated at the age of 12, 24, and 36 months. Prenatal exposure of 3.49 ng PAHs/m^3^ (less than 1 ng B[a]P/m^3^) affected the mental development index. Results suggest a possible risk of impairment in language, reading, and mathematics [5]. This cohort was later assessed by the Wechsler test at the age of five years [6] and at the age of 6–7 years by the CBCL method (Child Behavior Checklist) [7]. Prenatal PAH exposure affected verbal intelligence quotient score (IQ), and increased symptoms of anxiety, depression, and attention problems [7].

Peterson et al. [8] studied the impact of prenatal PAH exposure on the brain white matter and cognitive and behavioral functions using magnetic resonance imaging (MRI) on 40 children from the Columbia University cohort aged 7–9 years. They observed a reduced white matter surface of the left hemisphere of children with exposure to PAHs above median 8.2 ± 7.6 ng/m^3^, associated with slower information processing speed during intelligence testing, attention problems, and increased symptoms of Attention Deficit Hyperactivity Disorder (ADHD).

Another prospective cohort was followed in Krakow, Poland [9]. Children were assessed at the age of five years by the Raven test (Raven Colored Progressive Matrices, RCPM test). Prenatal exposure to PAHs higher than 17.96 ng/m^3^ decreased RCPM scores [10]. Transplacental exposures to PAHs were related to shorter head circumference, lower birth weight, and lower birth length, which may be later related to lower cognitive functions and poorer school performance [11].

These children in Krakow were further tested using the Wechsler test. Prenatal as well as postnatal PAH exposure decreased verbal IQ index. This was the first epidemiological study showing that prenatal PAH exposure measured as cord-blood PAH–DNA adducts was associated with cognitive dysfunction [12].

Air pollution by PAHs in Krakow is similar to the district of Karvina in the Czech Republic. This was the reason why we started to study the impact of air pollution on newborns in this district.

In the Czech Republic, the Moravian-Silesian Region is the most polluted region by PM2.5 (particulate matter < 2.5 μm) and c-PAHs (carcinogenic-PAHs) such as B[a]P. These are emitted by heavy industry and local heating systems. Accordingly, the impact of air pollution on newborns was studied in two districts: the more exposed district of Karvina (Moravian-Silesian Region, Northern Moravia) and the control district of Ceske Budejovice (Southern Bohemia) [13,14]. The study was very complex, analyzing the impact of air pollution by PAHs on genetic damage such as DNA adducts and gene expression, biomarkers of oxidative stress (8-oxodG adducts and lipid peroxidation), and concentration of OH–PAHs in the urine of mothers and newborns. c-PAHs bound to PM2.5 were collected by a High Volume Air Sampler (model ECO-HVS3000, Ecotech, Knoxfield, Australia) on Pallflex membrane filters (EMFAB, TX40HI20-WW) for three months during the period of collecting the biological samples [15].

Prenatal exposure to PAHs in cohorts of children from New York (USA) [5,6,7,8] and Krakow (Poland) [10,12] indicate the decrease in cognitive functions, intelligence quotient, and decrease in white matter volume in the left hemisphere.

As PAH concentrations in Krakow (Poland) correspond with PAH concentrations in Karvina (Czech Republic), we decided to study the impact of PAH exposure on children from Karvina and Ceske Budejovice (CB) during fetal development at the age of five years on their neurobehavioral functions. We tested the hypothesis that high concentrations of PAHs during prenatal development should affect neurobehavioral functions in the children.

## 2. Materials and Methods

### 2.1. Subjects

The cohorts were created in the summer of 2013 and winter of 2014 from newborns born at the Ceske Budejovice Hospital, Department of Obstetrics and Gynecology and Department of Neonatology and at the Karvina Hospital, Department of Obstetrics and Gynecology and Department of Neonatology. Newborns were selected from normal deliveries (38th–41st week) of non-smoking mothers who signed written consent. Cohorts included 99 newborns (summer) and 100 newborns (winter) in Ceske Budejovice, and 71 newborns (summer) and 74 newborns (winter) in Karvina. The study was approved by the Ethics Committee of both hospitals and the Institute of Experimental Medicine CAS in Prague.

Between November 2018 and November 2019, 199 mothers from the Ceske Budejovice district and 143 from the Karvina district who provided samples from their children in 2013 and 2014 were approached to take part in psychological testing. Undertaking the psychological test was optional. Out of the total number of 342 potential subjects, 140 refused to take part in the study, and 31 were impossible to contact. In the present study, data from 99 children from Ceske Budejovice and 70 children from Karvina were collected. The final sample therefore included 169 children.

This study was approved by the Faculty of Health and Social Sciences, University of South Bohemia, Ceske Budejovice. The Ethical Committee of the Faculty of Health and Social Science, University of South Bohemia from 30 June 2017.

### 2.2. Air Sampling and Analysis of Selected Air Pollutants

Particulate matter ≤2.5 µm (PM2.5) was collected by a High Volume (HiVol) 3000 Air Sampler (model ECO-HVS3000, Ecotech, Australia) on Pallflex membrane filters (EMFAB, TX40HI20-WW) in both study locations. The sampling was conducted as previously described [15]. Filters were collected each third day during the urine sampling. Detailed information on air sampling, extraction of organic complex mixtures (EOM) from the filters, and chemical analysis of B[a]P is described in Topinka et al. [15]. Concentrations of air pollutants were expressed in µg/m^3^ (PM2.5) and ng/m^3^ (B[a]P). Exposure to PM2.5 and B[a]P was calculated for each mother for her last trimester.

When data from HiVol samplers did not cover all trimesters, additional published data from the CHMI (Czech Hydrometeorologic Institute) were used [16]. The average daily concentrations of PM2.5 µm and benzo[a]pyrene were collected for both localities, Karvina and CB with the measure method–beta absorption for PM2.5 µm and gas chromatography with mass detection for B[a]P. Semi-individual exposure doses were assigned to individual mothers as averages of these values over the period based on the date of delivery.

### 2.3. Urine Samples, OH-PAHs (hydroxy-PAH metabolites) Detection

#### 2.3.1. Measurement of Urinary Creatinine

The creatinine values were used for normalizing the urine concentration/dilution in individual samples in order to ensure data comparability. The creatinine concentration was measured using a Jaffe spectrophotometric method according to our previous study [17]. In brief, a colored complex of creatinine with alkaline picrate was formed and subsequently measured at 505 nm.

#### 2.3.2. Analysis of 11 OH-PAHs in Urine

##### Extraction

The sample preparation procedure was based on liquid–liquid extraction (LLE) with the extraction solvent ethyl acetate and a clean-up step using dispersive solid-phase extraction (d-SPE) with a sorbent Z-Sep is described in detail in our previous paper [17].

##### Instrumental Analysis

The UHPLC–MS/MS (ultra high performance liquid chromatography - tandem mass spectrometry) analysis of 11 urinary OH-PAHs was performed using an Acquity Ultra-Performance LC system coupled to a triple quadrupole mass spectrometer Xevo TQ-S (both Waters, Milford, MA, USA) with electrospray ionization in a negative ion mode (ESI-). Analytes were separated on a PFP (pentafluorophenyl) Kinetex column, Phenomenex (USA) (100 mm × 2.1 mm × 1.7 μm). Measurement conditions are described in further detail in our previously published paper [17].

##### Quality Assurance/Quality Control and Validation

The validation of the analytical method for the analysis of 11 urinary OH-PAHs and the validation of Jaffe spectrophotometric method for the creatinine determination are described in detail in our previous study [18]. In each set of samples, the method accuracy was checked by using the Standard Reference Material (SRM) 3673 (Organic Contaminants in Non-Smokers’ Urine). Limits of quantification (LOQs) were in the range of 0.01–0.025 ng/mL with recoveries ranging between 77–113% and repeatability of 3–16%.

### 2.4. Measures of Child Visual-Motor Functioning and Intellect

To examine the potential effect of PAH exposure on cognitive development in five year old children, two psychological assessment instruments were used, namely the Bender Visual Motor Gestalt test and the Raven Colored Progressive Matrices test. From a variety of possible standardized tools, these two methods have been chosen bearing in mind the age of the tested children and the fact that measurements should be successfully done in one session. Both methods are well received by children and help them adapt to the test situation. Five year old children from our cohort were tested individually.

In order to assess the level of visual-motor functioning in five year old children, the Bender Visual Motor Gestalt test (BG test) was used. This test focuses on assessing motor functioning, visual perception, and potential developmental or neurological impairments in children and adults [18].

A total of 168 children at the age of five years completed the test. Each of our five year old children was presented with nine cards depicting different geometric shapes. The cards were presented individually and the tested children were asked to copy the design, trying to make the best reproduction possible. Test results were scored based on the organization and accuracy of the reproduction. This drawing test was well received by children and helped them considerably to get used to and feel comfortable with the test situation.

Once the BG test was completed, the children were presented with a non-verbal intelligence test called the Raven Colored Progressive Matrices (RCPM test) [19], which was also used in a similar study in Krakow, Poland [10]. The test has been developed and widely used for assessing reasoning and problem solving ability in children between five and 11 years including those suffering some kind of physical or mental impairment. The RCPM test consists of three sets of twelve matrix designs with increasing level of difficulty. A total of 167 children at the age of five years completed the RCPM test.

### 2.5. Questionnaire for Mothers

Mothers engaged in the study provided us with information regarding the social environment of the family, breastfeeding and eating habits, and child´s medical history. Similarly, the data regarding gestational age, birth weight, birth length, head circumference, and Apgar score were collected in order to be taken into account while analyzing the psychological test results.

### 2.6. Statistical Analysis

We used two statistical methods for the evaluation of differences in the cohorts. The Mann–Whitney U test (Wilcoxon rank-sum test) was used for direct comparison of the RCPM test and BG test results for PAH related values between cohorts.

Logistic regression was used for the purpose of estimating the impact of the type of delivery on the scores of the RCPM test and BG test as dependent values. Necessary conversion of rough scores of the test values into binary scale, values of OH-PAH metabolites, and EP PAHs values was done by dividing by medians of the appropriate group distribution.

The logistic regression quantifies impact intensity to calculated odds ratio (OddR), thus estimating the strength of the association between the independent when achieving the dependent testing score above the median of the group distribution [20].

Calculated OddRs in this analysis showed the probability with which children would achieve in the RCPM test and also the BG test scores above the median in their cohorts in association with the PAH exposure from environmental pollution represented either by OH–PAH metabolites in urine or by means of EP values above the median of its distribution.

For purpose of the exclusion of other possible confounders of estimated impacts, multiple other parameters were tested such as health and social status of mothers, mostly related to the maternal questionnaire like maternal age, maternal ETS (environmental tobacco smoke), various maternal health status parameters, children birth parameters and birth procedures, and quantified child illness by categories in the period from birth to two years of age. No other statistically significant impact was found. Impact of the type of delivery and the mother´s education was separately studied [21].

## 3. Results

Tested confounders are presented in Table 1. Comparing Karvina vs. Ceske Budejovice, in Karvina, mothers were younger, ETS exposure was higher during the first and second child year, gestation age was longer, birth length was shorter, Apgar 5´ was higher, TBC (tuberculosis) RJS primovaccination was higher, and gastrointestinal diseases in children were more frequent.

Results of the psychological tests in both districts are presented in Table 2. Neither outcome of the RCPM test nor the BG test differed between children in Karvina vs. Ceske Budejovice, or boys vs. girls.

Concentration of environmental pollution during the third trimester of mother´s pregnancy was calculated from regular pollution measurement according to concentrations determined within 90 days before delivery. Results significantly differed between Karvina and Ceske Budejovice: B[a]P 6.1 ± 4.53 vs. 1.19 ± 1.28 ng/m^3^, *p* < 0.001, PM2.5 37.7 ± 14.7 vs. 17.1 ± 4.8 µg/m^3^, *p* < 0.001 (Table 3).

All OH–PAH metabolites in the urine of children at the time of delivery were significantly higher in Karvina vs. Ceske Budejovice (with exception of 3-OH-B[a]P and 6-OH-chrysene) as they corresponded to a higher PAH concentrations in ambient air in Karvina (Table 4).When we analyzed the impact of B[a]P as well as PM2.5 exposure in ambient air during the last three months of pregnancy, we did not observe any statistically significant effect of the B[a]P exposure on the results obtained by the psychological tests, but the exposure to PM2.5 decreased the values of the BG test in Karvina in girls (OddR = 0.25, *p* < 0.05) (Table 5).

When we analyzed the relationship between the psychological test and PAH exposure, detected as OH-PAHs in urine at the time of delivery in the period 2013–2014, we did not observe any effects related to the OH–PAH metabolite values at the time of delivery and the results of the RCPM test and BG test in those children aged five years (Table 6).

When we analyzed the effect of confounders (Table 1), these confounders did not affect the results of psychological testing. When we analyzed the impact of these confounders on the results of the RCPM test and BG test, no effect of these confounders was observed.

A total of 61 mothers in our study had attained a university degree, while 87 had higher secondary and 17 lower secondary education. Interestingly, this ratio does not correspond to the values of mapping the level of education in the population according to the results of the population census conducted in 2011. A total of 36% of mothers in our study had achieved a university degree compared to the 12% listed by the census of the Czech population listed [22]. Ten percent of mothers willing to take part in their child´s psychological testing had attained a lower secondary education compared to 17.6% in the population according to the census. The mothers’ educational level significantly affected the results of the psychological tests (Table 7).

## 4. Discussion

Our results did not support our original hypothesis that high concentrations of PAHs in the ambient air in Karvina during prenatal development may affect cognitive functions in children at five years of age. This conclusion is surprising, as the concentration of B[a]P in Karvina during the third trimester of the mothers’ pregnancies was 6.1 ± 4.53 ng/m^3^. When we compared the OH–PAHs metabolites in the urine of newborns at the time of delivery, we did not find any effect of any OH–PAH metabolites on the cognitive development of five year old children. Surprisingly, increased concentrations of PM2.5 during the third trimester affected the results of the BG test in girls in Karvina. The rejection rate of part of the mothers in our cohorts represents a considerable limitation in our research.

Our results are in discrepancy with the results of studies by Perera et al. [5,6,7]. B[a]P concentrations in Karvina are at least five times higher than in New York. This difference may be partially related to the different ethnicity between African-Americans vs. Caucasians as well as social differences between those cohorts in the USA and the Czech Republic. As Lovasi et al. [23] already pointed out, child cognitive test scores in the Columbia University cohort were significantly affected by the neighborhood social context. The significant impact of low levels of antioxidants such as alpha-tocopherol, gamma-tocopherol, and carotenoid concentrations at age 6–9 years on neurodevelopment related to PAH prenatal exposure was also observed in the Polish cohort [24]. The quality of diet, vegetables, and fruit intake may be another reason for the discrepancy.

According to various studies, the heritability of intelligence is somewhere between 0.30–0.75 [25]; cognitive abilities can be influenced strongly by the environment, social enrichment, and way of upbringing [26] to name a few. It might be possible that mothers who were more interested in nurturing their children‘s neurodevelopment and cognitive abilities, thus compensating for the potential negative effects of the environment by increased care, were also those who were more willing to take part in our study as opposed to the number of those who refused the testing.

Edwards et al. [10] observed the effect of the prenatal exposure of airborne PAHs on five year old children in Krakow, Poland by also using the RCPM test, with the exposure to sum of PAHs was 17.96 ng/m^3^. This exposure was higher than the exposure in Karvina.

According to our previous study [21], the effect of the type of delivery, cesarean vs. vaginal, seems to be more significant in affecting cognitive functions in children than prenatal exposure to PAHs. Furthermore, we observed an important effect of mothers’ education level when comparing university vs. other education.

## 5. Conclusions

We studied the impact of PAH exposure in ambient air, determined in the ambient air as the concentration of benzo[a]pyrene (B[a]P) and OH–PAHs metabolites in the urine of newborns in the time of delivery on the cognitive development of five year old children using the Bender Visual Motor Gestalt Test (BG test) and the Raven Colored Progressive Matrices (RCPM test). We did not observe any effect of B[a]P exposure during the last trimester or OH–PAH metabolites at the time of delivery to cognitive development in 5 year old children. Higher exposure to PM2.5 during the third trimester in Karvina decreased the results of the BG test in girls in Karvina. We believe that the given topic deserves further research.

## Figures and Tables

**Table 1 brainsci-10-00619-t001:** Overview of tested confounders.

		All		CB		Karvina		Boys		Girls	
		*n*	Mean ± SD	*n*	Mean ± SD	*n*	Mean ± SD	*n*	Mean ± SD	*n*	Mean ± SD
Maternal Characteristics											
Maternal Age	years	168	31.9 ± 4.5	99	32.7 ± 4.4	69	30.8 ± 4.4 *	78	31.6 ± 4.0	90	32.2 ± 4.9
ETS—pregnancy	cig/day	167	0.06 ± 0.24	99	0.04 ± 0.20	68	0.09 ± 0.29	78	0.06 ± 0.25	89	0.06 ± 0.23
ETS—1st child year	cig/day	167	0.08 ± 0.27	99	0.03 ± 0.17	68	0.15 ± 0.36 **	78	0.09 ± 0.29	89	0.07 ± 0.25
ETS—2nd child year	cig/day	167	0.11 ± 0.32	99	0.06 ± 0.24	68	0.19 ± 0.40 **	78	0.12 ± 0.32	89	0.11 ± 0.32
Maternal University Education	%	167	0.37 ± 0.48	99	0.34 ± 0.48	68	0.41 ± 0.50	78	0.42 ± 0.50	89	0.33 ± 0.47
Birth Characteristics											
Vaginal Delivery	%	168	0.69 ± 0.46	99	0.70 ± 0.46	69	0.68 ± 0.47	78	0.69 ± 0.46	90	0.69 ± 0.47
Gestation Age	weeks	168	39.8 ± 1.8	99	39.5 ± 1.5	69	40.1 ± 2.0 ***	78	39.7 ± 1.3	90	39.9 ± 2.1
Birth Weight	G	162	3434 ± 439	97	3464 ± 452	65	3389 ± 417	76	3502 ± 439	86	3374 ± 432 ^++^
Birth Length	cm	159	49.7 ± 2.1	94	50.0 ± 1.9	65	49.2 ± 2.3 **	76	50.1 ± 2.1	83	49.3 ± 2.0 ^++^
Birth Head Perimeter	cm	158	34.4 ± 1.4	97	34.4 ± 1.5	61	34.4 ± 1.3	74	34.7 ± 1.5	84	34.2 ± 1.4
Apgar 5′		146	9.9 ± 0.5	88	9.8 ± 0.6	58	10.0 ± 0.1 **	66	9.8 ± 0.6	80	9.9 ± 0.3
Other Delivery Complication	%	168	0.05 ± 0.23	99	0.05 ± 0.22	69	0.06 ± 0.24	78	0.09 ± 0.29	90	0.02 ± 0.15
Hyperbilirubinia	%	168	0.09 ± 0.29	99	0.06 ± 0.24	69	0.13 ± 0.34	78	0.08 ± 0.27	90	0.10 ± 0.30
TBC Primovaccination	%	168	0.08 ± 0.27	99	0.04 ± 0.20	69	0.13 ± 0.34 *	78	0.05 ± 0.22	90	0.10 ± 0.30
Children’s Diseases											
GIS	count	168	0.32 ± 0.66	99	0.22 ± 0.56	69	0.45 ± 0.76 **	78	0.35 ± 0.75	90	0.29 ± 0.57
Viral Diseases	count	168	0.18 ± 0.43	99	0.22 ± 0.46	69	0.13 ± 0.38	78	0.22 ± 0.47	90	0.16 ± 0.39
Otitis	count	168	0.03 ± 0.20	99	0.03 ± 0.17	69	0.03 ± 0.24	78	0.04 ± 0.25	90	0.02 ± 0.15
HCD	count	168	0.29 ± 0.57	99	0.23 ± 0.53	69	0.36 ± 0.62	78	0.28 ± 0.53	90	0.29 ± 0.60
Bronchitis	count	168	2.47 ± 2.58	99	2.25 ± 2.42	69	2.78 ± 2.79	78	2.51 ± 2.68	90	2.43 ± 2.51

Results of Mann–Whitney U test compare by region * *p* < 0.05, ** *p* < 0.01, *** *p* < 0.001 and by gender ^++^
*p* < 0.01.

**Table 2 brainsci-10-00619-t002:** Results of the psychological tests.

	All		CB		Karvina		Boys		Girls	
	*n*	Mean ± SD	*n*	Mean ± SD	*n*	Mean ± SD	*n*	Mean ± SD	*n*	Mean ± SD
RCPM test	168	18.7 ± 4.6	99	18.9 ± 4.2	69	18.3 ± 5.0	78	18.7 ± 4.7	89	18.6 ± 4.6
BG test	169	32.5 ± 15.1	99	32.8 ± 14.5	70	32.1 ± 16.0	79	31.8 ± 16.3	89	33.1 ± 14.0

**Table 3 brainsci-10-00619-t003:** Concentration of environmental pollution in third trimester of mother’s pregnancies for delivery in the years 2013–2014.

	All		CB		Karvina		Boys		Girls	
	*n*	Mean ± SD	*n*	Mean ± SD	*n*	Mean ± SD	*n*	Mean ± SD	*n*	Mean ± SD
PM2.5 µm (µg/m^3^)	168	24.4 ± 13.3	99	17.1 ± 4.8	69	37.7 ± 14.7 ***	78	23.8 ± 13.2	90	24.8 ± 13.5
B[a]P (ng/m^3^)	168	3.18 ± 3.88	99	1.19 ± 1.28	69	6.1 ± 4.53 ***	78	3.08 ± 3.95	90	3.28 ± 3.85

Results of Mann Whitney U-test, compared by region*** *p* < 0.001.

**Table 4 brainsci-10-00619-t004:** Concentration of OH–PAHs in urine (µg/g creatinine) at the time of delivery in the years 2013–2014.

	All		CB		Karvina		Boys		Girls	
	*n*	Mean ± SD	*n*	Mean ± SD	*n*	Mean ± SD	*n*	Mean ± SD	*n*	Mean ± SD
1–OH–Naphthol	142	0.34 ± 0.47	87	0.11 ± 0.18	55	0.70 ± 0.56 ***	71	0.33 ± 0.41	71	0.34 ± 0.53
2–OH–Naphthol	142	3.89 ± 3.52	87	2.95 ± 2.38	55	5.37 ± 4.44 ***	71	3.69 ± 3.29	71	4.09 ± 3.75
2–OH–Fluoranthene	142	0.17 ± 0.15	87	0.11 ± 0.10	55	0.26 ± 0.16 ***	71	0.14 ± 0.14	71	0.19 ± 0.15 ^++^
1–OH–Phenantrene	142	0.42 ± 0.55	87	0.16 ± 0.16	55	0.84 ± 0.67 ***	71	0.48 ± 0.66	71	0.37 ± 0.40
2–OH–Phenantrene	142	0.23 ± 0.23	87	0.11 ± 0.11	55	0.42 ± 0.23 ***	71	0.20 ± 0.21	71	0.27 ± 0.24 ^++^
3–OH–Phenantrene	142	0.04 ± 0.05	87	0.02 ± 0.02	55	0.08 ± 0.06 ***	71	0.04 ± 0.05	71	0.04 ± 0.05
4–OH–Phenantrene	142	0.10 ± 0.32	87	0.06 ± 0.13	55	0.17 ± 0.48 ***	71	0.06 ± 0.14	71	0.14 ± 0.43
9–OH–Phenantrene	142	0.90 ± 1.66	87	0.36 ± 0.69	55	1.74 ± 2.29 ***	71	0.69 ± 1.40	71	1.10 ± 1.87
1–OH–Pyrene	142	0.07 ± 0.09	87	0.03 ± 0.04	55	0.13 ± 0.11 ***	71	0.06 ± 0.08	71	0.08 ± 0.10
6–OH–Chrysene	142	0.01 ± 0.00	87	0.01 ± 0.00	55	0.01 ± 0.00	71	0.01 ± 0.00	71	0.01 ± 0.00
3–OH–B[a]P	142	0.45 ± 0.00	87	0.45 ± 0.00	55	0.45 ± 0.00	71	0.45 ± 0.00	71	0.45 ± 0.00
Sum OH–PAH	142	6.14 ± 5.03	87	3.88 ± 2.90	55	9.71 ± 5.60 ***	71	5.67 ± 4.74	71	6.60 ± 5.29

Results of the Mann–Whitney U test, compared by region *** *p* < 0.001 and by gender ^++^
*p* < 0.01.

**Table 5 brainsci-10-00619-t005:** Estimated impact of environmental pollution in the third trimester of mothers’ pregnancies to psychological testing values.

			All		Boys			Girls		
		All	CB	Karvina	All	CB	Karvina	All	CB	Karvina
		OddR (95% CI)	OddR (95% CI)	OddR (95% CI)	OddR (95% CI)	OddR (95% CI)	OddR (95% CI)	OddR (95% CI)	OddR (95% CI)	OddR (95% CI)
RCPM test	PM2.5 µm, 3rd trim	0.62 (0.34–0.14)	0.96 (0.45–2.09)	0.76 (0.28–1.92)	0.63 (0.26–1.52)	1.09 (0.35–3.43)	0.90 (0.24–3.41)	0.59 (0.25–1.39)	0.88 (0.31–2.50)	0.64 (0.17–2.38)
	B[a]P, 3rd trim	0.62 (0.34–0.14)	0.71 (0.32–1.53)	0.67 (0.26–1.71)	0.77 (0.32–1.85)	0.92 (0.29–2.88)	0.90 (0.24–3.41)	0.49 (0.21–1.16)	0.57 (0.20–1.64)	0.51 (0.14–1.92)
BG test	PM2.5 µm, 3rd trim	0.63 (0.35–1.15)	0.52 (0.24–1.13)	0.42 (0.16–1.10)	0.90 (0.38–2.15)	0.78 (0.25–2.44)	0.71 (0.19–2.69)	0.44 (0.19–1.03)	0.37 (0.12–1.10)	0.25 (0.06–0.99) *
	B[a]P, 3rd trim	0.63 (0.35–1.15)	0.72 (0.33–1.57)	0.48 (0.19–1.23)	0.90 (0.38–2.15)	0.92 (0.29–2.88)	0.71 (0.19–2.69)	0.44 (0.19–1.03)	0.60 (0.20–1.75)	0.32 (0.08–1.24)

Logistic regression results * *p* < 0.05.

**Table 6 brainsci-10-00619-t006:** Impact of the 2013–2014 PAH–OH to the psychological testing values.

		All			Boys			Girls		
		All	CB	Karvina	All	CB	Karvina	All	CB	Karvina
		OddR (95% CI)	OddR (95% CI)	OddR (95% CI)	OddR (95% CI)	OddR (95% CI)	OddR (95% CI)	OddR (95% CI)	OddR (95% CI)	OddR (95% CI)
RCPM test	Sum PAH–OH	1.26 (0.68–2.34)	0.97 (0.44–2.11)	0.87 (0.34–2.25)	1.62 (0.66–3.96)	1.54 (0.47–5.02)	0.46 (0.12–1.83)	0.97 (0.41–2.30)	0.66 (0.23–1.89)	1.78 (0.45–6.97)
	2–OH–Naphthol	1.04 (0.56–1.92)	1.05 (0.48–2.29)	1.10 (0.43–2.86)	1.21 (0.50–2.94)	1.07 (0.33–3.47)	0.75 (0.19–2.92)	0.87 (0.37–2.04)	1.02 (0.35–2.91)	1.40 (0.35–5.54)
BG test	Sum PAH–OH	1.80 (0.97–3.31)	0.86 (0.39–1.89)	0.96 (0.37–2.47)	1.80 (0.74–4.36)	0.56 (0.23–2.43)	0.56 (0.14–2.21)	1.73 (0.74–4.07)	0.91 (0.31–2.64)	1.78 (0.45–6.97)
	2–OH–Naphthol	1.34 (0.73–2.46)	0.80 (0.37–1.76)	1.21 (0.47–3.14)	1.09 (0.45–2.62)	0.75 (0.23–2.43)	0.56 (0.14–2.21)	1.56 (0.67–3.65)	1.70 (0.28–2.34)	2.31 (0.57–9.40)

**Table 7 brainsci-10-00619-t007:** Results of the psychological tests and the mother’s education level.

Mother’s Education Level										
	All		CB		Karvina		Boys		Girls	
Lower secondary	*n*	Mean ± SD	*n*	Mean ± SD	*n*	Mean ± SD	*n*	Mean ± SD	*n*	Mean ± SD
RCPM test	16	15.0 ± 2.7 ***	5	15.6 ± 2.9 **	11	14.7 ± 2.8 ***	9	15.0 ± 3.4 ***	7	15.0 ± 1.8 ***
BG test	17	24.1 ± 12.9 **	5	25.0 ± 15.7	12	23.8 ± 12.2 **	10	23.7 ± 15.0 *	7	24.7 ± 10.2
Higher secondary	*n*		*n*		*n*		*n*		*n*	
RCPM test	51	18.1 ± 4.4 ***	30	18.4 ± 4.3 *	21	17.4 ± 4.7 **	24	18.0 ± 5.0 *	27	18.1 ± 4.1 *
BG test	51	31.8 ± 14.3	30	32.4 ± 14.3	21	30.5 ± 14.6	24	29.9 ± 17.4	27	32.9 ± 12.1
University	*n*		*n*		*n*		*n*		*n*	
RCPM test	51	20.6 ± 4.5	30	20.4 ± 3.9	21	20.7 ± 5.2	24	20.5 ± 4.0	27	20.6 ± 5.1
BG test	51	35.2 ± 16.0	30	34.5 ± 14.7	21	36.1 ± 17.6	24	35.1 ± 14.5	27	35.3 ± 17.8

Results of the Mann–Whitney U test compared by type of mother’s education related to university level degree * *p* < 0.05, ** *p* < 0.01, *** *p* < 0.001.

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
