# Peer review of "Effect of Polycyclic Aromatic Hydrocarbons Exposure on Cognitive Development in 5 Years Old Children"

_brainsci, 2020, doi:10.3390/brainsci10090619_

Round 1
Reviewer 1 Report
The authors have addressed all my previous comments. I only have one minor revision to suggest: in Table 3, please also present the 95% CIs in addition to ORs.
Author Response
Table 3 is now Table 5, 95% CI was added as proposed, also in Table 6.
Reviewer 2 Report
Major comments
- Previous comments were not thoroughly addressed.
- No information on exposure assessment or description of exposure metrics used was provided in the Methods sections.
- Statistical analysis methods are flawed.
- The manuscript still contains many grammatical errors.
General comments
Abstract
Line #22 Results: should report the results of PAH exposure measures.
Methods
Line#105-135 please describe exposure metrics and justify the validity of the metrics.
Line#168 “Analogicaly has been transformed values of OH-PAHs metabolites and EP PAHs values during 3rd trimester.” – the meaning of this sentence is unclear.
Line #172-178 No need to explain Odds ratio.
Line #184 “the mother´s education was separately studied” - parental education is a known potential confounder of the outcomes of interest and thus should be examined in the analysis.
Results
Line # 186 – Cohort characteristics should be presented first in the results section.
Line# 188-190 all information should be self-contained. It is unclear how “Concentration of environmental pollution during the third trimester of mother´s pregnancy” were estimated. Are they (B[a]P and PM2.5) one time single measurements or averages of multiple measurement over a period of time? Are they 1-hr or 24-hr averages or integrated over a even longer time period? More details on the distributions of exposure variables should be provided.
Line #191- differences in exposure variables between groups should be reported before reporting the impact of exposure on the outcomes.
Author Response
Thank you for your comments, the text was substantially changed.
Enclosed you will find the text with marked changes.
General comments
Abstract
Line #22 Results: should report the results of PAH exposure measures.
added
Methods
Line#105-135 please describe exposure metrics and justify the validity of the metrics.
added
Line#168 “Analogicaly has been transformed values of OH-PAHs metabolites and EP PAHs values during 3rd trimester.” – the meaning of this sentence is unclear.
deleted
Line #172-178 No need to explain Odds ratio.
deleted
Line #184 “the mother´s education was separately studied” - parental education is a known potential confounder of the outcomes of interest and thus should be examined in the analysis.
Information about maternal education was added as Table 7.
Results
Line # 186 – Cohort characteristics should be presented first in the results section.
Changed as proposed, now Table 1.
Line# 188-190 all information should be self-contained. It is unclear how “Concentration of environmental pollution during the third trimester of mother´s pregnancy” were estimated. Are they (B[a]P and PM2.5) one time single measurements or averages of multiple measurement over a period of time? Are they 1-hr or 24-hr averages or integrated over a even longer time period? More details on the distributions of exposure variables should be provided.
added, now l. 190-192
Line #191- differences in exposure variables between groups should be reported before reporting the impact of exposure on the outcomes.

This manuscript is a resubmission of an earlier submission. The following is a list of the peer review reports and author responses from that submission.
Round 1
Reviewer 1 Report
This study assessed the assocition between PAH exposures and cognitive development in 5 years old children. While the topic is interesting, the manuscript was not well written, and the analyses were flawed in many ways. My main concerns include 1) the selection bias as it is unclear how the children were sampled and the loss-to-follow-up proportion is high, 2) the small sample size, which may be underpowered to detect the association and a formal power calculation is needed, and 3) the lack of control for potential confounders. Below are my detailed comments.
1. Introduction: the introduction is too long and can be shortened. The descriptions on previous studies are too detailed, which should be summarized. Instead of listing these details, more efforts should be made to discuss the limitations in previous studies and then clearly state the rationale to conduct this study: for example, what gap this study will bridge, and what added value this study will bring.
2. Introduction, page 2, line 63: should be "Saunders et al."
3. Methods, page 3, line 130: how were newborns selected? Randomly? or just as a convenience sample?
4. Methods, page 5, line 193: do you mean "We used two ..."? There needs to be a seperate section describing the potential confounders and how they were selected. Currently it seems no confounder is adjusted. In addition, why dichotomize BG test score based on the median? Why not model the score directly or convert it to a binary variable based on a clinically meaning cut point?
5. Table 1, page 5: in additoin to the results of psychological tests, please also include the distributions of covariates.
6. Table 2, page 6: concentration of OH-PAHs is likely to be skewed, and therefore needs to be log-transformed. Instead of using mean, geometric mean should be calculated.
Reviewer 2 Report
The paper of Blazkova et al. reports on relationships between airborne PAH concentrations and children cognitive development in Karvina (polluted area) and Ceske Budejovice (control area) in Czech. The idea is very relevant, but unfortunately was not properly implemented.
General comments
The purpose of the paper was “to study the impact of PAHs exposure during fetal development” on children’s neurobehavioral functions as measured at age 5. The hypothesis was that exposure to PAHs during perinatal (prenatal?) period affects neurobehavioral development in children. OH-PAHs measured at age 5 (in 2018-2019) are irrelevant to prenatal exposure.
The short-term nature of the exposure measure, OH-PAHs in urine samples, fundamentally limited the study’s ability to evaluate potential associations between PAH exposure and children developmental effects. Urinary metabolites of PAHs, such as OH-PAHs, reflect exposures in past one or two days at the most. It is inappropriate to use OH-PAHs in one-time spot urine samples to represent prenatal exposure or any semi-chronic to chronic exposures. As reported in the paper, the sum OH-PAH almost doubled from 2013/2014 to 2018/2019 in both regions, indicating the time-varying nature of the exposure and thus the less usefulness of one-time measurement.
Secondly, OH-PAH are not specific to inhalation exposure. Instead, it is known to be substantively influenced by dietary PAH intakes. Therefore, they can’t be directly used as biomarkers of exposure to airborne PAHs without a careful examination of other exposure pathways. Without an appropriate exposure measure, it is impossible to investigate the associations between exposure and health effects.
Authors did not detect any differences in health outcome measures (RCPM or BG test) between children in Karvina and in Ceske Budejovice. One reason could be exposure misclassification. While the Sum OH-PAH and many individual OH-PAHs significantly differed between two regions, individual OH-PAH concentrations had large variations within each group. Group comparison strategy did not allow establishing exposure/response relationship when the within-group-variations are larger than the between-group-variations. This is probably why the authors saw significant effects of 2-OH-naphthol exposure on RCPM within the groups while no difference between groups were observed. In this case, conclusions on the relationship can be misleading. Refinements in exposure assessment and statistical analysis are needed to improve the analysis.
Specific comments
Introduction – literature review should summarize what is known, what the data gaps are, and what new information this study may bring.
Line#44-48, animal studies and SO2 exposure is not relevant to this paper.
Line#49-71, the studies in Teplice indicated the correlation between PAH concentrations in ambient air and NES2 test scores. There was no evidence support the statements of “Obtained results proved for the first time the significant impact of air pollution in the mining districts of Northern Bohemia to neuropsychological development in children,” nor “higher c-PAHs concentrations 61 could significantly affect neuropsychological development in children in the mining districts.”
Line #106, PAH-DNA adducts could potentially be used to assess PAH exposures, which have longer half-lives than urinary metabolites.
Method – Statistical analysis was not clearly described. It is unclear how confounding factors (e.g., social-economic status, other chemical exposures, 2nd hand smoking, alcoholic of parents…) will be controlled. This needs to be described.
Results – should describe the characteristics of the study population first, and then followed by the results of exposure. These should be reported before presenting results of health outcomes (Table 1). A thorough analysis on exposure and variations between- and within-group is needed.
Line 214-222, contradictory statements, e.g. “…analyzed the relationship between psychological test and PAHs exposure, detected as 214 OH-PAHs in urine (Table 2), we did not observe any effect of exposure in the period 2013/4 as well 215 as 2018/9” and “RCPM test was significantly affected by the sum of 2-OH-naphthol exposure in 2018/9 …” Again, OH-PAHs measured in 2018-2019 has nothing to do with the prenatal exposure.
Discussion -
Line#263-274, should be in Results - characteristics of the study population.